# Risk mapping using serologic surveillance for selected One Health and transboundary diseases in Cambodian goats

Jarunee Siengsanan-Lamont[1], Lida Kong[1,2], Theng Heng[1,2], Sokun Khoeun[2], Sothyra Tum[2], Paul W. Selleck[1], Laurence J. Gleeson[1], Stuart D. Blacksell[1,3]*

**1** Mahidol-Oxford Tropical Medicine Research Unit, Faculty of Tropical Medicine, Mahidol University, Bangkok, Thailand, **2** National Animal Health and Production Research Institute (NAHPRI), General Directorate of Animal Health and Production (GDAHP), Phnom Penh, Cambodia, **3** Centre for Tropical Medicine, Nuffield Department of Clinical Medicine, John Radcliffe Hospital, Oxford, United Kingdom

* stuart.blacksell@ndm.ox.ac.uk, stuart@tropmedres.ac

**Data Availability Statement:** All data generated or analysed during this study included in this

## Abstract

In Cambodia, goat production and meat consumption are customary among Muslim communities. Recently, goat meat has gained popularity among Cambodians. Goat farmers use a traditional management system, including grazing, requiring minimal labour. The close proximity between humans and animals could increase the risk of zoonotic disease transmission. A serological survey was undertaken to estimate the prevalence of some priority zoonoses and high-impact animal diseases in the Cambodian goat population. A total of 540 samples were collected from goats in six provinces and analysed with commercially available enzyme-linked immunosorbent assays for *Brucella* species, Q fever (*Coxiella burnetii*), Foot and Mouth Disease virus non-structural protein (FMDV NSP) and Peste des Petits Ruminants virus (PPRV). True seroprevalences with a 95% Confidence Interval (CI), taking into account imperfect tests, risk factors and odds ratios (ORs), were calculated to better understand the disease distribution and epidemiology. Independent variables used in statistical modellings included sex, body condition score, age, vaccination history, province and commune, while dependent variables were ELISA test results. The overall true prevalence of antibodies to *Brucella* spp., *C. burnetii*, FMDV and PPRV, were 0.1% (95% CI 0.0, 1.0), 7.2% (95% CI 5.3, 9.7), 57.7% (95% CI 53.1, 62.3) and 0.0% (95% CI 0.0, 0.0), respectively. There was no identified risk factor for brucellosis and PPR. The two risk factors for *C. burnetii* seropositivity were sex (p-value = 0.0005) and commune (p-value <0.0001). However, only the OR of *C. burnetii* seropositive female goat was significant at 9.7 (95% CI 2.7, 35.5) times higher than male. The risk factors of FMD NSP seropositivity were age (p-value = 0.001) and commune (p-value <0.0001). Only the age 'more than two-year-old' group with a significant OR of 6.2 (95% CI 2.1, 18.4) using the 'up to one-year-old' group as the reference. In summary, *Brucella spp.* seroprevalence was low, while no evidence of PPRV antibodies was detected in the goat populations. *C. burnetii* seroprevalence in female goats was significantly higher than for males, and there were significant differences in *C. burnetii* seroprevalence between communes. The overall FMDV NSP seroprevalence was high, especially in older animals. Vaccination should be advocated to protect animals from FMDV

publication are available at the Open Science Framework at https://osf.io/ztrcf/?view_only=b3b392faefc4478eb46086b8ecf9da85.

**Funding:** This research was funded by the Biological Threat Reduction Program (BTRP) of the Defense Threat Reduction Agency (DTRA) of the US government [contract number HDTRA1-08-D-0007](JS-L, LK, TH, SK, ST, PS, LG). This research is supported in part by the Wellcome Trust [220211] of the United Kingdom (SB). The funders had no role in study design, data collection and analysis, decision to publish, or preparation of the manuscript.

**Competing interests:** I have read the journal's policy and the authors of this manuscript have the following competing interests: SB is a Section Editor at PLoS Neglected Tropical Diseases.

and improve productivity. As the impacts of these zoonoses on human and animal health were still unknown, further investigation of these zoonotic diseases' epidemiology is recommended.

## Author summary

Goat production and meat consumption are growing in Cambodia. Smallholders often keep goats in a housing area within their households, increasing contact between animals and humans and, thereby, the risk for zoonotic disease transmission. This study collected 540 goat samples applying the two-stage survey method, where the communes were the sampling frame and villages were clustered and tested for antibodies against *Brucella spp.*, *Coxiella burnetii* (Q fever), Foot and mouth disease virus non-structural protein (FMDV NSP), and Peste des Petits Ruminants virus (PPRV) using enzyme-linked immunosorbent assay (ELISA). The overall apparent seroprevalence of *Brucella*, *C. burnetii*, FMDV NSP and PPRV were 0.4%, 7.2%, 53.3% and 0.9% (n = 540), respectively. The only two brucella-positive samples were from Kampot and Tboung Khmum provinces. Moreover, Kandal province (22.2%, N = 90) and Dei Edth commune (36.7%, N = 30) had the highest *C. burnetii* seroprevalence compared to the others. Although seroprevalence of *Brucella spp.* detected in our study was low, while *C. burnetii* seroprevalence in some studied areas was relatively high, and the impacts of both zoonoses on animal and human health remain unknown. Seroprevalence of FMDV NSF was also high and widely distributed. Thus, future investigations targeting high-risk areas would be recommended to understand better the disease epidemiologies and their impacts on human and animal health.

## Introduction

In 2018, the total number of goats in Cambodia reported by the General Directorate of Animal Health and Production (GDAHP) was 25,747 head [1]. There is a strong cultural demand for sheep and goat meat in the country, and the Observatory of Economic Complexity (OEC) reported that in 2019, Cambodia imported around $2.9 million worth of sheep and goat meat, while there was virtually no export of this meat [2]. Some government and international organisations have given goats to farmers to boost household incomes [3,4]. While goats provide meat for family consumption, they are also considered an additional source of income for the family. Cambodian farmers raise goats using a traditional management system of releasing them for grazing during the daytime and housing them at night [5]. Goats are commonly kept for consumption by the Cham Muslim minority. However, recently, goat meat has been increasingly consumed by other Cambodians [6], and there is an emerging market opportunity for goat meat in the country, as local production of other meat (beef, pork, chicken and duck) is still in short supply [7]. Goat production can help improve small farmers' livelihoods and provide alternative red meat for Cambodians.

Livestock health is directly related to human health. Animal production could increase the risks of zoonotic disease transmission, especially in low-resource environments where disease surveillance and response systems are not well established [8]. Smallholders usually keep their livestock within their households, which allows close contact between humans and animals. Moreover, One Health pathogens reported in goats in neighbouring countries that could impact Cambodian goat farmers include *Coxiella burnetii* (Q fever), *Brucella melitensis* [9],

*Brucella spp.* [10], endoparasite [11] and protozoan [12] infections. Other high-impact animal diseases, including Foot and Mouth Disease (FMD) [13, 14] and orf infection [15], were also reported. Some of these diseases are considered endemic in Cambodia. However, available information on goat population, management, health and production in Cambodia is limited [16]. Previous studies of Cambodian goats mainly focused on forages, nutrition, and parasite burdens and control [5, 17–19]. There were no publications on goat disease surveillance programs at the time of our study. Thus, this study's main objectives were to enhance the animal disease diagnostic capacity through practising, estimate the prevalence of priority diseases in Cambodian goats (including FMD, brucellosis and Q fever), and provide scientific evidence of Peste des Petits Ruminants (PPR) freedom. The outcomes of this study will contribute to future disease control and prevention programs which would help improve goat productivity and national food security, as well as provide information to inform public health about potential zoonosis exposure in the human population.

## Material and methods

### Ethics statement

Animal ethics approval for this survey no. 5074GDAHP, was obtained from the General Directorate of Animal Health and Production, Cambodia.

### Site Selection and sample size calculation

Goat populations in 2018 reported the by General Directorate of Animal Health and Production (GDAHP) [1] were used as baseline information (Fig 1). A phone interview with all 25 Cambodian veterinary provincial chiefs in late 2019 revealed that only nine provinces, namely Kandal, Takeo, Kampot, Tboung Khmum, Prey Veng, Phnom Penh, Kampong Chhnang, Siem Reap and Battambang, raised goats. The remaining provinces reported that farmers no longer raised goats or had no goat production in the area. The selection criteria of the targeted provinces and districts were based on 1) high numbers of goats and 2) high activity of goat meat consumption, while communes and villages were randomly selected.

Six provinces, comprising Kandal, Takeo, Kampot, Tbung Khmum, Prey Veng and Phnom Penh, were selected for this study. Although there was a high population of goats in Battambang in 2018, the phone assessment indicated a significant reduction in numbers. The Epitools ("2-Stage surveys for demonstration of freedom") [20] was used to calculate the sample size. The two-stage sample size calculation used communes as the sampling frame and villages as clusters. The parameters used for the sample size calculation included a survey system sensitivity of 95%, unit-level design prevalence of 10%, cluster-level design prevalence of 1%, test sensitivity of 95% (based on the ELISA test sensitivity used), and target cluster sensitivity of 95%.

The numbers of goats kept in those randomly selected villages ranging from 20 to 200 (except a village with 350 goats) were used in the calculation. Thus, the sample size of 30 goats per village (cluster) was selected for this study. For those villages with less than 30 goats, all goats were to be sampled. However, some randomly selected villages had no goats during the sample collection period. Thus, some communes and villages included in this study were based on available goat population and convenient access to replace those without goats. Locations of the study sites are shown in Table 1 and visualised in Fig 2.

### Sample collection and transportation

Animal information, including age, sex, vaccination history and body condition score (BCS), was collected using a sample collection form accompanying animal specimens. The BCS used for

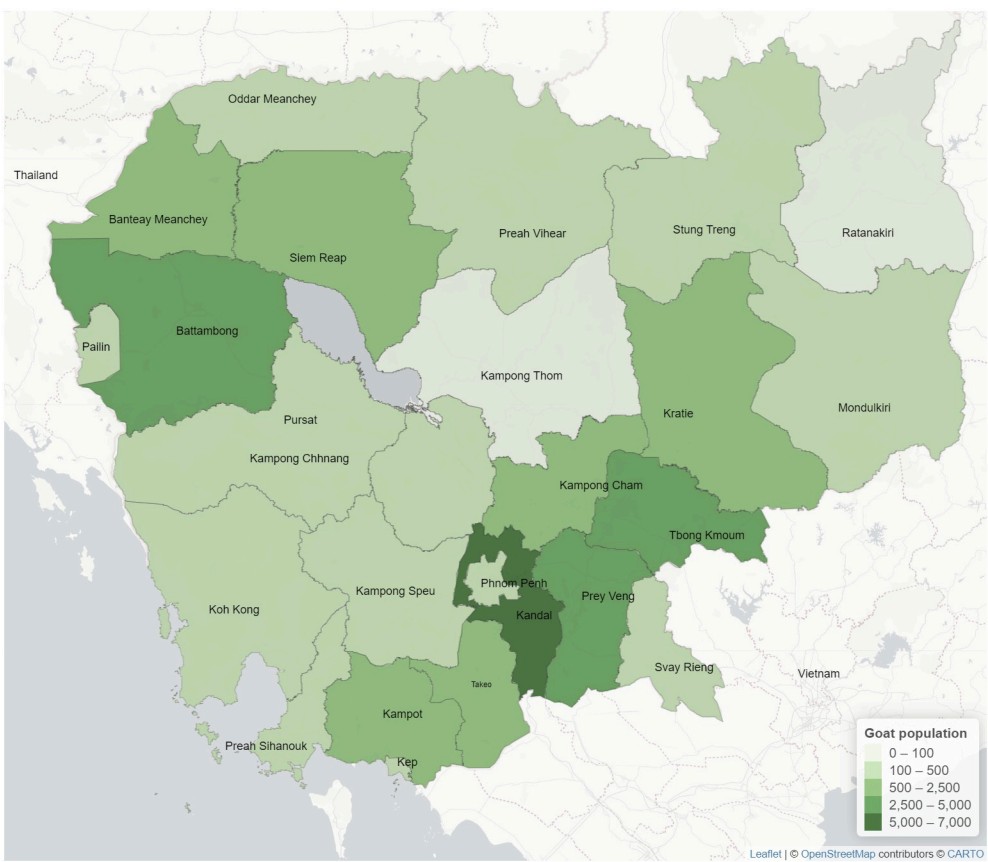

**Fig 1. Density map indicating Cambodian goat population in 2018.** Map shapefiles sourced from https://www.diva-gis.org/gdata.

our study was the score 1 (lean) to 5 (fat). A blood sample was collected from the jugular vein of the goat and transferred to a sterile with no anticoagulant (red-top) tube. The tube was labelled with a unique identification number using a permanent marker and placed in a rack inside a large zip-lock plastic bag. The samples were kept in a cooler with ice packs when transported back to the National Animal Health and Production Research Institute (NAHPRI) laboratory. Serum was separated from whole blood at NAHPRI using a refrigerated centrifuge (Thermo Scientific, Germany) at 1000–2000 x g for 10 minutes and stored in an -30°C Freezer until tested.

## Laboratory and statistical analyses

Serum samples were tested for antibodies specific to FMDV non-structural protein (NSP), *Coxiella burnetii*, *Brucella spp*. and PPRV using enzyme-linked immunosorbent assay (ELISA) commercial kits by ID Vet, (310 rue Louis Pasteur, 34790 Grabels, France, https://www.id-vet. com/) [ID Screen FMDV 3ABC NSP competition ELISA (Cat# FMDNSPC-5P), ID Screen brucellosis serum indirect multi-species (Cat# BRUS-MS-10P; detecting antibodies against *B. abortus*, *B. melitensis* or *B. suis*), ID Screen Q fever indirect multi-species (Cat# FQS-MS-5P), and ID Screen PPRV Competition (Cat# PPRC-4P)]. Laboratory testing was conducted at NAHPRI based on the manufacturer's instructions, and results were determined by an ELISA microplate reader (Infinite F50, Singapore). The ID soft software version 5.05 [21] provided by the manufacturer was used for the calculation of the test sample percentage of positive control (S/P%) or the test sample percentage of negative control (S/N%), depending on the test. The

**Table 1. The sample size of each selected village.**

| Province | District | Commune | Village | Total sample collected |
|---|---|---|---|---|
| Kampot | Banteay Meas (BM) | Tuk Meas Khang Lech | Tuk Meas Khang Lech | 30 |
| | Dang Tong (DT.) | Srae Chea Khang Tboung | Tropeang Sda | 30 |
| | Kampong Trach (KT) | Damnak Kantuot Khang Cheung | Phnom Domrey | 30 |
| Kandal | Kien Svay (KS) | Dei Edth | Dei Edth | 30 |
| | Mukh Kampul (MK.) | Svay Ampear | Thmey | 30 |
| | Ponhea Lueu (PL.) | Kampong Os | Kampong Os | 7 |
| | | | Ta Proun | 23 |
| Phnom Penh | Ruessei Kaev (RK) | Chrouy Changvar | Chrouy Changvar | 30 |
| | Sen Sok (SS) | Preak Pnov | Preak Pnov | 30 |
| | Pou Senchey (PSc) | Trapeang Krasang | Chaom Chau | 30 |
| Prey Veng | Bar Phnum (BP) | Cheung Phnum | Svay Samsib | 30 |
| | Peam Chor (PC) | Svay Pluoh | Sammaki | 30 |
| | Preah Sdach (PS.) | Lvea | Kongpong Thnoul | 30 |
| Takeo | Angkor Borei (AB.) | Preaek Phtoul | Angkor Borei | 30 |
| | Borei Cholsar (BC) | Borei Cholsar | Kampong Ompil | 30 |
| | Tram Kak (TK) | Cheang Tong | Ang Riem | 30 |
| Tboung Khmum | Krong Suong (Ksu) | Sangkat Suong | Vihea Loung | 29 |
| | Ou Reang Ov (ORO) | Preah Theat | Phum 44 | 30 |
| | Tboung Khmum (TKh) | Chirou Ti Muy | Chirou Ti Muy | 31 |
| | | | **Total** | **540** |

diagnostic test sensitivity (Dse), specificity (Dsp) and the cutoff of the S/P% for FMDV NSP, brucellosis and Q fever ELISAs were previously reported by Siengsanan-Lamont [22, 23]. For the PPRV ELISA test, the competition percentage (S/N%) was calculated, and the cutoff was applied as recommended by the manufacturer, i.e. S/N% ≤ 50% = positive, 50%< S/N% ≤60% = doubtful, S/N% > 60% = negative. An earlier study reported a Dsp of 97.1% and a Dse of 96.4% sensitivity for the PPRV ELISA [24].

Descriptive and statistical analyses were performed in Microsoft Excel and R studio version 4.1.0 [25] (R Core Team, 2021). Apparent and true prevalence with a 95% confidence interval (95%CI) were calculated, taking into account an imperfect test using the EpiR package, Wilson method [26]. Seroprevalence maps were produced using the R leaflet package [27]. Univariate analyses (Fisher Exact and Chi-squared tests) were used to identify associations between the variables and test results [28]. The independent variables used in modelling included sex, BCS, age, vaccination history, province, and commune, while the dependent variables were test results. Subsets of variables with a p-value less than 0.1 were included in multivariate logistic regression modelling [29]. The final model contained significant variables (p-value < 0.05) by the analysis of variance (ANOVA), Chi-squared test, smallest Akaike Information Criterion (AIC) among fitted models [30], no multicollinearity (variance inflation factor (VIF) <10) [31], and passed the Hosmer-Lemeshow, goodness of fit test [32]. Odds ratios (OR) with 95% CI of the final model's variables were then calculated.

## Results

### The survey was performed between October and December 2020

A total of 540 goat samples were collected from nineteen villages within eighteen communes and districts due to one village having a small number of goats (Table 1). The plot of age

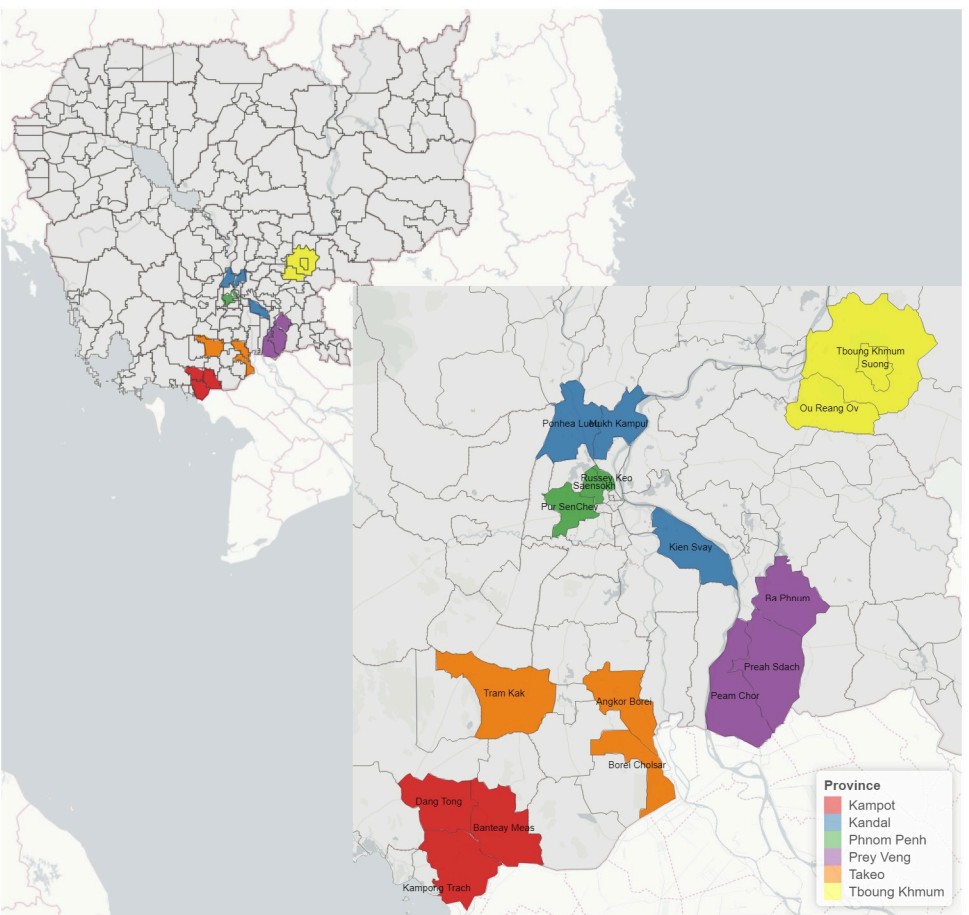

**Fig 2. Sample collection locations in Cambodia.** Map shapefiles sourced from https://www.diva-gis.org/gdata.

(month) versus the sex variable is shown in Fig 3. Animal ages were then grouped into three groups, as described in Table 2, for statistical analyses and model-fitting purposes. More than 70% (381/540) of the samples were from female goats, while approximately 68% (368/540) were aged 'up to one-year-old' (Table 2). Based on BCS scoring 1 to 5, 64.3% (347/540) were from animals with a BCS of 3. However, only 22.2% (120/540) were vaccinated against FMDV or both FMDV and haemorrhagic septicaemia (HS). All FMDV-vaccinated animals (n = 30) were from Angkor Borei village, while the FMDV and HS-vaccinated animals were from three villages [Chrouy Changvar (n = 30), Kampong Ompil (n = 30) and Ang Riem(n = 30)]. The overall true seroprevalence of FMDV NSP, *Brucella spp*. and *C. burnetii* were 57.7%, 0.1% and 7.2% (n = 540), respectively. For PPRV, despite the apparent seroprevalence of 0.9%, taking into account the diagnostic test sensitivity and specificity, the true prevalence was zero. The overall apparent and true seroprevalences are shown in Table 2.

Visualisations of true seroprevalence per province are presented in Fig 4. The provinces with the highest seroprevalence for FMDV NSP and *C. burnetii* were Kampot and Kandal, respectively. Only two samples were *Brucella* positive, one from Kampot and one from Tboung Khmum (Fig 4). All of the goat samples from three communes (Borei Cholsar, Cheung Phnum and Tuk Meas Khang Lech) were positive for FMD NSPV antibodies. In contrast, all samples collected from Ang Riem village, Cheang Tong commune, Takeo province, were negative in the FMDV NSP ELISA.

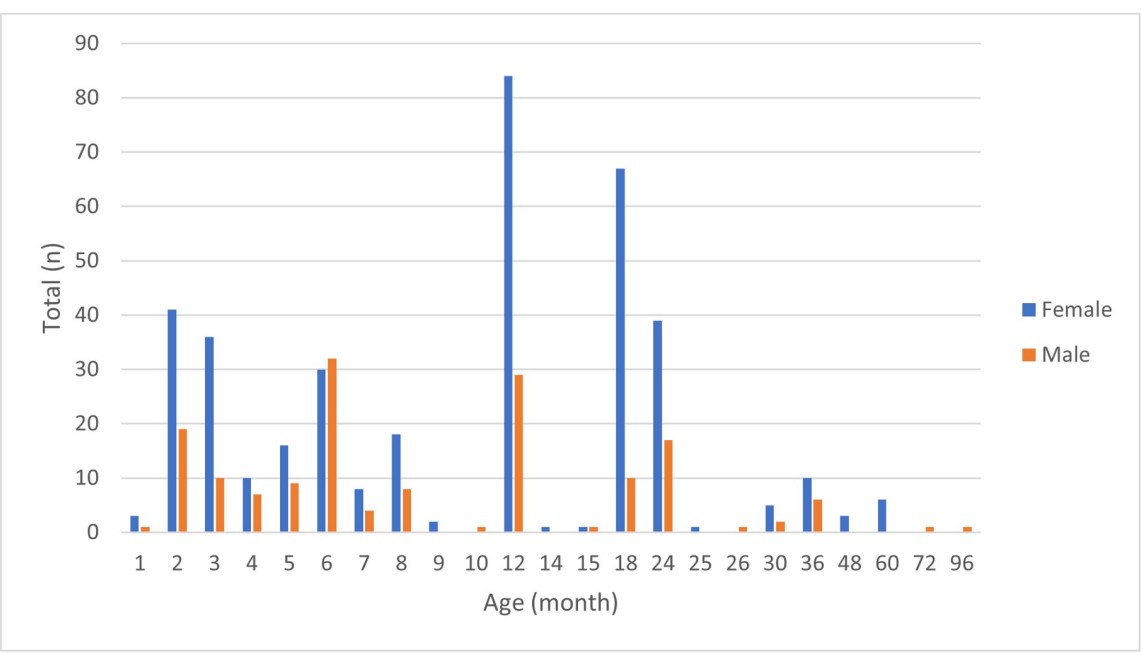

**Fig 3. Total numbers of females and males and age (months).**

For FMDV NSP (Table 3), risk factors identified by univariate analyses were age (p-value = 0.001), vaccination history (p-value <0.0001), province (p-value <0.0001) and commune (p-value <0.0001). These variables were included in the multivariate logistic regression modelling where the final model contained only age (p-value = 0.001) and commune (p-value <0.0001). Compared to the 'up-to-one-year' age group, the 'more than two-years-old' group had a significantly higher FMDV NSP seropositivity (OR = 6.2, 95%CI 2.1, 18.4). The significant factors identified by univariate analyses for *C. burnetii* were sex (p-value = 0.004), province (p-value <0.0001) and commune (p-value <0.0001). Only sex (p-value = 0.0005) and commune (p-value <0.0001) remained in the final multivariate model. However, only the OR of female goats (9.7 (95% CI 2.7, 35.5)) compared to males was significant, while all ORs of communes were not significant. None of the variables was significantly associated with *Brucella* and PPRV seropositivity.

## Discussion

A report on the goat population between 1998 and 2008 by the Food and Agriculture Organization (FAO) demonstrated no goat raising or meat production in Cambodia [33]. In contrast, a study in 2004 funded by the Australian Centre for International Agricultural Research (ACIAR) claimed the Cambodian goat population was approximately 5000 head [16]. As the GDAPH official report [1] on goat populations in 2018 was not up to date, data obtained from the 2019 phone interview was used for the sample size calculation. Despite the 2018 government report (Fig 1) showing that almost all provinces had goats, the phone interview confirmed that only nine provinces still had goats. Goats kept by Cambodian households were classified into three groups: smallholders with 3–10 goats, medium with 10–49 goats and large with more than 49 goats [16] with an average number of goats per household of 21.5 [34]. The numbers of goats in the randomly selected village (between 20–200 goats) derived from the phone interview were used for the sample size calculation with a target of 30 samples per

**Table 2. Seroprevalences taken into account imperfect tests (True prevalence).**

| Variable | Total (n) | FMDV NSP positive | % FMDV NSP seroprevalence (95% CI) | Brucellosis positive | % *Brucella spp.* seroprevalence (95% CI) | Q fever positive | % *C. burnetii* seroprevalence (95% CI) | PPRV positive | % PPRV seroprevalence (95% CI) |
|---|---|---|---|---|---|---|---|---|---|
| *Sex* | | | | | | | | | |
| - F | 381 | 211 | 60.0 (54.4, 65.4) | 0 | 0.0 (0.0, 0.7) | 36 | 9.4 (6.9, 12.8) | 5 | 0.0 (0.0, 0.1) |
| - M | 159 | 77 | 52.3 (43.9, 60.8) | 2 | 1.0 (0.0, 4.2) | 3 | 1.9 (0.6, 5.4) | 0 | 0.0 (0.0,0.0) |
| *Age (month)* | | | | | | | | | |
| - <= 12 | 368 | 212 | 62.4 (56.8, 67.9) | 2 | 0.2 (0.0, 1.7) | 23 | 6.3 (4.2, 9.2) | 3 | 0.0 (0.0,0.0) |
| - 13–24 | 136 | 54 | 42.7 (34.0, 51.9) | 0 | 0.0 (0.0, 2.5) | 12 | 8.8 (5.1,14.8) | 2 | 0.0 (0.0, 2.5) |
| - >24 | 36 | 22 | 66.3 (48.4, 81.8) | 0 | 0.0 (0.0, 9.3) | 4 | 11.1 (4.4, 25.3) | 0 | 0.0 (0.0, 7.2) |
| *BCS (1–5)* | | | | | | | | | |
| - 2 | 188 | 94 | 54.0 (46.2, 61.8) | 1 | 0.2 (0.0, 2.7) | 8 | 4.3 (2.2, 8.2) | 2 | 0.0 (0.0, 1.0) |
| - 3 | 347 | 190 | 59.3 (53.5, 65.0) | 1 | 0.0 (0.0, 1.3) | 30 | 8.7 (6.1, 12.1) | 3 | 0.0 (0.0,0.0) |
| - 4 | 5 | 4 | 87.1 (40.3, 100.0) | 0 | 0.0 (0.0, 43.3) | 1 | 20.0 (1.0, 62.4) | 0 | 0.0 (0.0, 43.4) |
| *Vaccine history* | | | | | | | | | |
| - FMDV | 30 | 29 | 100.0 (90.8, 100.0) | 0 | 0.0 (0.0, 11.1) | 0 | 0.0 (0.0, 11.4) | 0 | 0.0 (0.0, 9.0) |
| - FMDV-HS | 90 | 37 | 44.2 (33.6, 55.6) | 0 | 0.0 (0.0, 3.8) | 5 | 5.6 (2.4, 12.4) | 0 | 0.0 (0.0, 1.3) |
| - No vaccine | 420 | 222 | 57.2 (51.9, 62.4) | 2 | 0.2 (0.0, 1.4) | 34 | 8.1 (5.9, 11.1) | 5 | 0.0 (0.0,0.0) |
| *Province* | | | | | | | | | |
| - Kampot | 90 | 72 | 87.1 (76.7, 94.8) | 1 | 0.8(0.0, 5.8) | 8 | 8.9 (4.6, 16.6) | 0 | 0.0 (0.0, 1.3) |
| - Kandal | 90 | 44 | 52.8 (41.7, 64.0) | 0 | 0.0 (0.0, 3.8) | 20 | 22.2 (14.9, 31.8) | 2 | 0.0 (0.0, 5.2) |
| - Phnom Penh | 90 | 16 | 18.5 (11.3, 28.6) | 0 | 0.0 (0.0, 3.8) | 9 | 10.0 (5.4, 17.9) | 0 | 0.0 (0.0, 1.3) |
| - Prey Veng | 90 | 47 | 56.5 (45.2, 67.5) | 0 | 0.0 (0.0, 3.8) | 0 | 0.0 (0.0, 4.1) | 1 | 0.0 (0.0, 3.4) |
| - Takeo | 90 | 59 | 71.2 (59.8, 81.1) | 0 | 0.0 (0.0, 3.8) | 0 | 0.0 (0.0, 4.1) | 0 | 0.0 (0.0, 1.3) |
| - Tboung Khmum | 90 | 50 | 60.1 (48.8, 71.0) | 1 | 0.8 (0.0, 5.8) | 2 | 2.2 (0.6, 7.7) | 2 | 0.0 (0.0, 5.2) |
| *Commune* | | | | | | | | | |
| - Borei Cholsar | 30 | 30 | 100.0 (96.6, 100.0) | 0 | 0.0 (0.0, 11.1) | 0 | 0.0 (0.0, 11.4) | 0 | 0.0 (0.0, 9.0) |
| - Chirou Ti Muy | 31 | 15 | 52.2 (34.1, 70.7) | 0 | 0.0 (0.0, 10.8) | 1 | 3.2 (0.2, 16.2) | 0 | 0.0(0.0, 8.7) |
| - Cheang Tong | 30 | 0 | 0.0 (0.0, 11.4) | 0 | 0.0 (0.0, 11.1) | 0 | 0.0 (0.0, 11.4) | 0 | 0.0 (0.0, 9.0) |
| - Chrouy Changvar | 30 | 7 | 24.6 (11.9, 44.0) | 0 | 0.0 (0.0, 11.1) | 5 | 16.7 (7.3, 33.6) | 0 | 0.0 (0.0, 9.0) |
| - Sangkat Suong | 29 | 27 | 100.0 (84.9, 100.0) | 0 | 0.0 (0.0, 11.4) | 1 | 3.5 (0.2, 17.2) | 1 | 0.6 (0.0, 15.3) |
| - Damnak Kantuot Khang Cheung | 30 | 20 | 72.4 (52.7, 87.9) | 0 | 0.0 (0.0, 11.1) | 1 | 3.3 (0.2, 16.7) | 0 | 0.0 (0.0, 9.0) |
| - Cheung Phnum | 30 | 30 | 100.0 (96.6, 100.0) | 0 | 0.0 (0.0, 11.1) | 0 | 0.0 (0.0, 11.4) | 0 | 0.0 (0.0, 9.0) |
| - Kampong Os | 30 | 3 | 9.9 (2.7, 27.1) | 0 | 0.0 (0.0, 11.1) | 1 | 3.3 (0.2, 16.7) | 1 | 0.5 (0.0, 14.7) |
| - Dei Edth | 30 | 28 | 100.0 (85.6, 100.0) | 0 | 0.0 (0.0, 11.1) | 11 | 36.7 (21.9, 54.5) | 0 | 0.0 (0.0, 9.0) |
| - Lvea | 30 | 7 | 24.6 (11.9, 44.0) | 0 | 0.0 (0.0, 11.1) | 0 | 0.0 (0.0, 11.4) | 1 | 0.5 (0.0, 14.7) |
| - Trepeang Krasang | 30 | 8 | 28.3 (14.5, 47.9) | 0 | 0.0 (0.0, 11.1) | 4 | 13.3 (5.3, 29.7) | 0 | 0.0 (0.0, 9.0) |
| - Preah Theat | 30 | 8 | 28.3 (14.5, 47.9) | 1 | 3.0 (0.1, 16.4) | 0 | 0.0 (0.0, 11.4) | 1 | 0.5 (0.0, 14.7) |
| - Preaek Phtoul | 30 | 29 | 100.0 (90.8, 100.0) | 0 | 0.0 (0.0, 11.1) | 0 | 0.0 (0.0, 11.4) | 0 | 0.0 (0.0, 9.0) |
| - Preak Pnov | 30 | 1 | 2.6 (0.0, 17.3) | 0 | 0.0 (0.0, 11.1) | 0 | 0.0 (0.0, 11.4) | 0 | 0.0 (0.0, 9.0) |
| - Srae Chea Khang Tboung | 30 | 22 | 79.8 (60.1, 93.5) | 1 | 3.0 (0.0, 16.4) | 1 | 3.3 (0.2, 16.7) | 0 | 0.0 (0.0, 9.0) |
| - Svay Ampear | 30 | 13 | 46.7 (29.1, 65.9) | 0 | 0.0 (0.0, 11.1) | 8 | 26.7 (14.2, 44.4) | 1 | 0.5 (0.0, 14.7) |

*(Continued)*

**Table 2.** (Continued)

| Variable | Total (n) | FMDV NSP positive | % FMDV NSP seroprevalence (95% CI) | Brucellosis positive | % *Brucella spp.* seroprevalence (95% CI) | Q fever positive | % *C. burnetii* seroprevalence (95% CI) | PPRV positive | % PPRV seroprevalence (95% CI) |
|---|---|---|---|---|---|---|---|---|---|
| - Svay Pluoh | 30 | 10 | 35.6 (20.1, 55.4) | 0 | 0.0 (0.0, 11.1) | 0 | 0.0 (0.0, 11.4) | 0 | 0.0 (0.0, 9.0) |
| - Tuk Meas Khang Lech | 30 | 30 | 100.0 (96.6, 100.0) | 0 | 0.0 (0.0, 11.1) | 6 | 20.0 (9.5, 37.3) | 0 | 0.0 (0.0, 9.0) |
| Overall apparent prevalence | 540 | 288 | 53.3% (49.1, 57.5) | 2 | 0.4% (0.1, 1.3) | 39 | 7.2% (5.3, 9.7) | 5 | 0.9% (0.4, 2.1) |
| Overall true prevalence | 540 | 288 | 57.7% (53.1, 62.3) | 2 | 0.1% (0.0, 1.0) | 39 | 7.2% (5.3, 9.7) | 5 | 0.0% (0.0,0.0) |

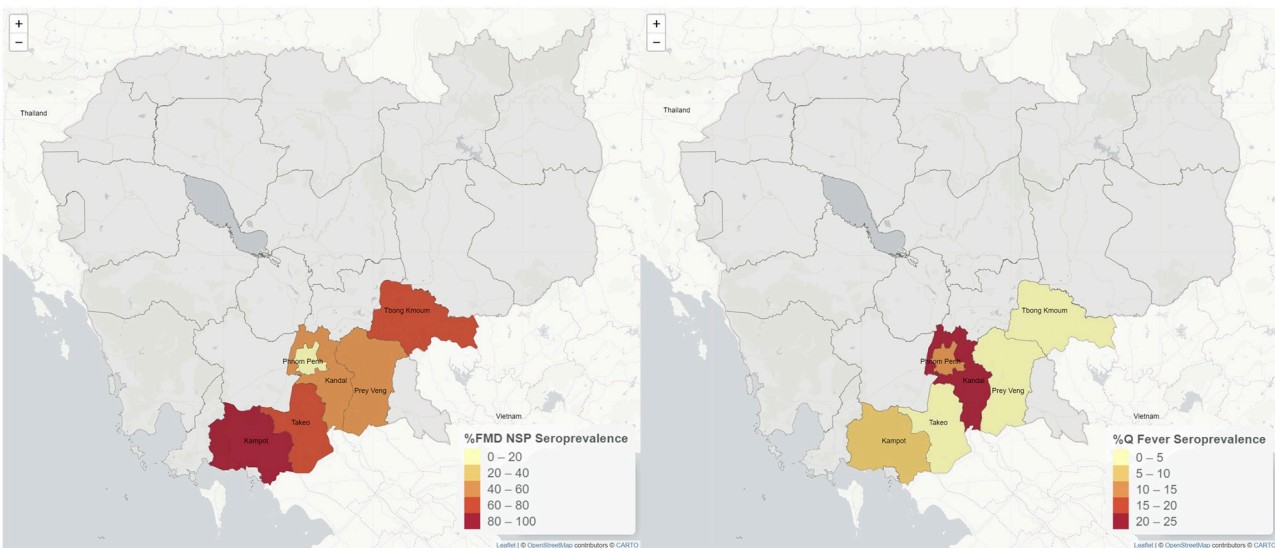

**Fig 4. True FMDV NSP and Q Fever seroprevalence at the provincial level.** Map shapefiles sourced from https://www.diva-gis.org/gdata.

**Table 3. Statistical modelling of FMDV and C. burnetii.**

| Dataset | Predictive variable (reference group) | p-value (Chi-Square) |
|---|---|---|
| FMDV NSP | *Univariate analysis* | |
| | - Age (less than 12 months) | 0.001 |
| | - Vaccination history (FMDV vaccine) | <0.0001 |
| | - Province (Kampot) | <0.0001 |
| | - Commune (Borei Cholsar) | <0.0001 |
| | *Multivariate logistic regression* | |
| | - Age (less than 12 months) | 0.001 |
| | - More than 24 months; OR 6.2 (2.1, 18.4) | |
| | - Commune (Borei Cholsar) | <0.0001 |
| C. burnetii | *Univariate analysis* | |
| | - Sex (Male) | 0.004 |
| | - Province (Kampot) | <0.0001 |
| | - Commune (Borei Cholsar) | <0.0001 |
| | *Multivariate logistic regression* | |
| | - Sex (Male) | 0.0005 |
| | - Female; OR 9.7 (2.7, 35.5) | |
| | - Commune (Borei Cholsar) | <0.0001 |

village. However, the sample collections conducted about a year after the phone interview revealed that the goat populations were again drastically changed and did not represent the 2020 goat distribution. Thus, not all communes and villages included in the study were randomly selected. The parameters used for the sample size calculation were selected according to the Dse of the kit (for test sensitivity) and a level of confidence at 95% (for survey system and cluster sensitivities). As for the design prevalences, the unit-level prevalence of 10% and the cluster-level prevalence of 1% were sufficient to detect disease evidence in the target population. Moreover, due to limited resources, our study did not include Kampong Chhnang and the north-western provinces (Battambong and Siem Reap), even though significant goat populations were confirmed. It is recommended that future programs should include these areas, especially as these provinces are in close proximity to neighbouring countries.

Our data revealed that almost half of the sampled goats were females aged less than or equal to 12 months. Goats are generally sold for meat aged between 3 and 5 months (11–23 kg) [35]. Moreover, goats reach a breeding age between 5 and 13 months of age [36]. A study in Vietnam also reported that goat farmers commonly kept one male for breeding and sold other males at a live weight of 25–30 kgs [36]. This practice may explain the lower numbers of males in almost all age-month groups compared to females (except for the 6-month group) in our study. Despite the traditional management (mainly grazing) being common practice and most animals in our study being unvaccinated, most animals had a BCS of 3, considered to indicate adequate nutrition (not too thin or too fat). Only four (in Takeo and Phnom Penh provinces) of 18 villages vaccinated their goats against FMDV or FMDV/HS. It was unclear why the vaccines were only used in limited areas.

Seroprevalence of brucellosis in Cambodian goats was lower than those of studies in goats in Lao PDR (1.4%, n = 1458) [9] (Burns et al., 2018) and Thailand (1.4%, n = 94722) [37]. In contrast, *C. burnetii* seroprevalence of goats in Cambodia was higher than those reported in Lao PDR (4.1%, n = 1458) [9] (Burns et al., 2018) and Thailand (3.5%, n = 516) [38]. While for cattle, seroprevalences of *Brucella spp.* and *C. burnetii* in Cambodia (n = 477) were 0.2% and 0.8%, respectively [39]. For FMDV NSP, surprisingly, Cambodian goats in our study had a much higher seroprevalence (57.7%) than Lao PDR goats (13.0%, n = 591) [14]. A study reported that 11.1% of the goat population in outbreak subdistricts in Thailand was affected by FMD [40]. In conclusion, the seroprevalence of *Brucella spp.* in Cambodian ruminants was relatively low. Further investigation of brucellosis in the areas where seropositive animals were detected would be useful for future disease control.

The seroprevalence of *C. burnetii* in female goats was significantly higher than in males. *C. Burnetii* causes reproductive losses (abortion and stillborn) in females where the bacteria are highly concentrated in the body and amniotic fluids of aborting animals. As most male goats in our study were age less than or up to 12 months old, they may be less likely to be exposed to the bacteria during their lifetime compared to older female goats. As *C. burnetii* seroprevalence in Cambodian goats was higher than in other animals and in the neighbouring countries [40,41] and it is a One Health pathogen, the disease's impacts on animal and human health should be further investigated.

Purified FMDV vaccines are free from cellular proteins and NSP. Thus, antibodies against FMDV NSP are only present in animals previously exposed to FMDV (natural infection). However, the FMDV NSP antibodies can also be detected if NSP-contaminated vaccines have been used. FMD seroprevalence in the '1 to 2-year-old' group is a good indication of the endemic nature of the disease, as this group is unlikely to have received multiple vaccinations or have maternally derived antibodies. However, for this age group, a seroprevalence of 42.7% is very high in our study. The highest FMDV NSP seroprevalence was the 'more than two years old' group, followed by the 'up to one-year-old' group. This may be due to the older the

animals are, the more likely the animals are to be vaccinated multiple times (increasing chances of exposure to an unpurified vaccine) or exposed to FMD viruses, as FMD is endemic in Southeast Asia [22]. The vaccination history was also a risk factor for FMDV NSP seropositivity in this study. However, the seroprevalence of FMDV-vaccinated animals was much higher than those of unvaccinated and of FMDV/HS-vaccinated groups. However, all FMDV-vaccinated animals were from one village. Thus, it is likely that the village had been affected by an FMD outbreak. Communes were an FMD risk factor as all samples from some communes tested positive. In contrast, some other communes had low or no positive samples, which resulted in significant differences in FMDV seropositivity between communes. Goats in our study had a higher FMDV seroprevalence than other species in the country [42] and those in neighbouring countries [43]. Future studies investigating risk and protective factors within the communes would help support FMD control and prevention.

Our study revealed that PPRV true seroprevalence and 95% CIs were zero or approaching zero. PPR is a severe viral disease with morbidity and mortality rates of up to 90% and 40%, respectively [44]. As there was no previous report of PPR in the country prior to the time of this study, our study provided scientific evidence to support PPR freedom. However, Thailand reported the first detection of PPR in goats imported from West Africa on February 2021 [45]. More studies should be conducted to monitor the extent of the PPR introduction to the Southeast Asia region.

In conclusion, knowledge of diseases and their epidemiology is key to maximise goat productivity and protect human health. Some of the pathogens appear to be circulating in the population based on the seroprevalence results, except PPR. As the seroprevalence of FMDV NSP was high, and most goats in our study were not vaccinated, the use of vaccination which will help reduce losses and increase productivity, should be advocated. Seroprevalences of the two zoonoses, brucellosis and Q fever, were relatively low, although further investigation would help better understand the epidemiology and human health impacts of both neglected zoonotic diseases.

## Acknowledgments

This surveillance of goat diseases was a component of the enhancement of zoonotic disease outbreak detection in the Lao PDR and Cambodia (CAMNN3) project implemented in Cambodia by the National Animal Health and Production Research Institute (NAHPRI), General Directorate of Animal Health and Production, and the Mahidol-Oxford Tropical Medicine Research Unit (MORU). The authors are grateful for the efforts of the field and laboratory staff at NAHPRI.

## Author Contributions

**Conceptualization:** Jarunee Siengsanan-Lamont, Lida Kong, Sokun Khoeun, Sothyra Tum, Laurence J. Gleeson, Stuart D. Blacksell.

**Data curation:** Jarunee Siengsanan-Lamont, Lida Kong, Theng Heng.

**Formal analysis:** Jarunee Siengsanan-Lamont, Lida Kong, Theng Heng.

**Funding acquisition:** Stuart D. Blacksell.

**Investigation:** Jarunee Siengsanan-Lamont, Lida Kong, Theng Heng, Paul W. Selleck.

**Methodology:** Jarunee Siengsanan-Lamont, Lida Kong.

**Project administration:** Jarunee Siengsanan-Lamont, Lida Kong, Sokun Khoeun, Sothyra Tum, Laurence J. Gleeson, Stuart D. Blacksell.

**Supervision:** Sokun Khoeun, Paul W. Selleck, Laurence J. Gleeson, Stuart D. Blacksell.

**Writing – original draft:** Jarunee Siengsanan-Lamont, Stuart D. Blacksell.

**Writing – review & editing:** Lida Kong, Sothyra Tum, Laurence J. Gleeson, Stuart D. Blacksell.

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
