## [Decision Letter · Decision Letter 0]

7 Dec 2022

Dear Dr. Blacksell,

Thank you very much for submitting your manuscript "Risk mapping using serologic surveillance for selected One Health and transboundary diseases in Cambodian goats" for consideration at PLOS Neglected Tropical Diseases. As with all papers reviewed by the journal, your manuscript was reviewed by members of the editorial board and by several independent reviewers. In light of the reviews (below this email), we would like to invite the resubmission of a significantly-revised version that takes into account the reviewers' comments. 

We cannot make any decision about publication until we have seen the revised manuscript and your response to the reviewers' comments. Your revised manuscript is also likely to be sent to reviewers for further evaluation.

Sincerely,

Brett M. Forshey

Academic Editor

Dileepa Ediriweera

Section Editor

In this manuscript, Siengsanan-Lamont and co-authors describe the seroprevalence of antibodies against a variety of pathogens in goats in Cambodia. The reviewers felt that the study was largely well conducted and solid scientifically. There were a number of points they raised that need to be addressed prior to publication: 

- There is not sufficient description of study limitation in the Discussion section.

- The Methods are lacking some key details about the ELISAs used in the study, such as the sensitivity and specificity and the specific antigenic targets.

- The authors should describe seroprevalence in terms of antibodies against the pathogens, not in terms of disease (e.g. PPRV not PPR; C. burnetii not Q fever; etc).

- The authors should address any impact of FMDV vaccination on the results.

- The Abstract is much longer than it needs to be. I would put this one as optional, but I think it would be a stronger publication if the Abstract were more concise and clearer. I would say the same about the Author Summary.

- The Methods do not describe how the specific goat herds were selected, and I don't believe there was any statistical adjustment for clustering.

- Data availability, per PLOS policy, is not sufficiently addressed. There is no compelling statement on why these data can't be shared, and just mentioning to ask a government agency without even providing contact information does not address the requirement. 

Reviewer 1 also had a number of relevant and substantive comments in a MS Word attachment. Please make sure you address these as well - they should be considered carefully in your modifications. 

Some other minor comments:

- Abstract, line 32: "odds" in "odds ratios" does not need to be capitalized, and probably best described as "calculated" or "estimated" not "identified"

- Abstract, line 35: Looks like there is a typo in the confidence interval for FMD NSP.

- Abstract: I don't understand this statement "Q fever seroprevalence of female goats was significantly higher than males; however, it may be due to an imbalance between male and female populations." What "imbalance"? Would this have been addressed by statistical adjustments?

- Abstract: Why is FMD identified by the NSP when the other pathogens the antigen is not mentioned?

- Author Summary, lines 59-60: The framing of "true seroprevalence" is confusing. 

- Similarly in the Results, line 180: How can a true prevalence be "less than zero"? I think that's part of the problem I have with this interpretation. So, the assumption is that the few positives are false-positives, which may very well be true, it's just hard to accept outright that the 95% CI is bounded by 0.0 when there were samples that tested positive.

- Discussion, line 277: "Our study provided scientific evidence to support PPR freedom." Is this meant to convey that they study supports the idea that these regions are free of PPRV? I understand that the tool you used is not sufficiently specific, but then those 5 positives need to be better characterized if you are trying to demonstrate absence. I think it's fair say that PPRV is low to non-existent but more specific assays are needed, or something less concrete that is consistent with the data.

- Discussion, lines 268-271: The statement about the future of goat production is unrelated to the current study. I recommend you keep your final conclusions tied to the current study.

Reviewer's Responses to Questions

**Key Review Criteria Required for Acceptance?**

**Methods**

-Are the objectives of the study clearly articulated with a clear testable hypothesis stated?

-Is the study design appropriate to address the stated objectives?

-Is the population clearly described and appropriate for the hypothesis being tested?

-Is the sample size sufficient to ensure adequate power to address the hypothesis being tested?

-Were correct statistical analysis used to support conclusions?

-Are there concerns about ethical or regulatory requirements being met?

Reviewer #1: The objective is clearly stated, and the study design, while in need of more information, is appropriate. Population and sample size are well described, and while I am no statistician, it seems like the statistical analysis has been done correctly. I have attached a word document with more comments.

Reviewer #2: This is a well-conducted study with clear results which are well reported. There are no obvious scientific flaws in the work.

Line 136-7. Centrifugal force should always be expressed as "x g", not "revolutions per minute", as the force will vary according to the diameter of the centrifuge rotor as well as the rotational speed.

Reviewer #3: The objective of the study is clearly articulated. The study design is appropriate to address the stated objectives. The study population is clearly described and appropriate for the hypothesis being tested. The sample size is sufficient to ensure adequate power to address the hypothesis being tested. The statistical analyses used are inadequate to support conclusions. Ethical and regulatory requirements have been met.

L116-117: Are you demonstrating sero-prevalence of the diseases or freedom from the diseases/ the two objectives result in very different sample sizes. Also, you were investigating four diseases. The limits of the parameters used for sample size calculation were for which lead disease and tests? And why?

It is better if the figures and the table are separated by the sections of text on pg. 3 as appropriate. For instance, have Figure 1 immediately after explaining what it is about, then continue with text and place the table and Figure 2 after the second bit of text.

L117-118: It is not clear what this sentence means. Also state clearly what your sampling method was;

L116-122: Give the formula that Epitools utilized and explain the parameters in the formula followed by the limits you used (which you have described) and why. Give citations/references as well;

L121-122….sample size of 30 goats per village was calculated….But then according to Table 1, some villages had sample sizes <30;

L122-123: place this sentence in the results section;

L135: But direct contact with icepacks could cause haemolysis of whole blood. How did you prevent this?;

L154: These cutoffs have some overlap;

L155: These parameters/limits are more important in the section where you calculate the sample size;

L157/58: Give the formula used, the parameters used and their limits, why and the appropriate references. As in 116-122, even if a tool is used, there is an underlying formula;

L163: Given that you have more or less used cluster sampling technique, A Generalized Linear Mixed regression Model (GLMM) would have been more appropriate for your analysis so as to include random and fixed effects;

In Figure 2, show where Cambodia is located globally then zoom out the study areas, making clear the link between the names in the legend and those in the map and text;

**Results**

-Does the analysis presented match the analysis plan?

-Are the results clearly and completely presented?

-Are the figures (Tables, Images) of sufficient quality for clarity?

Reviewer #1: The results are a bit difficult to follow and I therefore recommend that the authors separate them by pathogen. In general the figures are good, but I am missing a table that presents results from the predictor variable analysis (it might just be that I missed it, in which case I do apologize). I have attached a word document with more comments.

Reviewer #2: Yes, with the exception of Table 1 which I found unclear as to which "Districts/Municipalities" belonged to which "Province". Maybe the demarcations should be clearer with the Province" name at the very top of the column.

Line 180. The true prevalence cannot be "less than zero" ! It is zero.

I wonder if Figure 4 (%Brucellosis seroprevalence) has any validity with only 2 positive samples in the whole study ! I would leave it out of the paper.

Reviewer #3: You could present results on herd structure (age, sex) and husbandry characteristics (vaccination, grazing) of the study animals before presenting the test results) to justify some of the discussions in the discussion section;

Instead of n=540; n=30, you should indicate how many out of the total i.e. the fraction forming the percentage e.g. 70% (378/540) in L172;

L172/73: Say …more than 70% of samples were from female animals;

You can break the narrative at line 181, present the table and proceed; 

L182: Figure 3 is not about sero-prevalence in the provinces. Correct and place the figure after L182;

Present Figure 4 at line 187 and mention it in the text;

L196: Rephrase to avoid two brackets into each other;

L236: Retain the citation format prescribed by the journal, throughout the manuscript;

Where the p values are very small, e.g. 8.0x10-7 in L189 etc., indicate them as p<0.001

**Conclusions**

-Are the conclusions supported by the data presented?

-Are the limitations of analysis clearly described?

-Do the authors discuss how these data can be helpful to advance our understanding of the topic under study?

-Is public health relevance addressed?

Reviewer #1: The conclusions are in general supported by the data, but the limitations need to be better described. Public health relevance is addressed. I have attached a word document with more comments.

Reviewer #2: Yes.

Reviewer #3:

L235-253: Do not just repeat your results and compare them to those of neighbouring countries. Discuss what the implications are;

L257-259: Is irrelevant;

L268-271: Not a conclusion from your results;

L273: You have no results indicating that most goats were not vaccinated and therefore cannot discuss

**Editorial and Data Presentation Modifications?**

Reviewer #1: Please see attached document.

Reviewer #2: All points covered above.

Reviewer #3: The detailed suggestions are listed for each section:

Abstract

L23: …..‘close proximity’;

L28: ….were used to detect……;

L31/32: Risk factors were identified using odds ratios;

L31-33: categorize the variables into dependent and independent variables;

L34: Say Brucella species rather than Brucella spp.;

L35: There is possibly an error in the CI for 7.2%;

L37 and throughout the manuscript: when the p-value is very small, it is expressed as

 p<0.001 or p<0.0001 depending on the significance levels desired;

L39: Only age, more than…. years was a significant risk factor……;

L39-41: For FMD NSP, why isn’t commune a significant risk factor when p=2.2x10-16 or rephrase what you have said to make it clear;

L42/43: Correct to: Q fever seroprevalence for female goats was significantly higher than for males. Why would an imbalance between male and female populations affect the sero-prevalence?-see also L266-267;

L49: It shouldn’t be that animals should be vaccinated on evidence from sero-prevalence. Evidence of actual disease signs or virus detection is better proof of disease presence;

L…- L…isn’t the author summary the same as the abstract?;

Introduction

L78: Replace ‘nowadays’ with ‘currently’;

L80: …as local production is still low (remove still in short supply);

L86: Moreover, some health pathogens reported……..;

L99-100: Include PPR since you investigated it as well;

L103: Put a full stop at the end of the sentence;

**Summary and General Comments**

Reviewer #1: My main comments are that the manuscript sometimes is a bit confusing for the reader, and I have listed some structural comments in the attached document. I also think that the study design description lacks important information and that the discussion needs work, and that aspects such as study limitations should be better included.

Reviewer #2: Good study worthy of publication.

Reviewer #3: The authors have indicated that in Cambodia, goat production and goat meat consumption are customary among Muslim communities. Recently, goat meat has gained popularity among Cambodians. Goat farmers use a traditional management system, including grazing, requiring minimal labour. Some government and international organisations have promoted goat production as a secondary source of income for small animal holders to improve their livelihoods. The close proximately between humans and animals could increase the risk of zoonotic disease transmission. A serological survey was undertaken to estimate the prevalence of some priority zoonoses and high-impact animal diseases in the Cambodian goat population. This is an important study that is relevant to the journal. The results are relevant in highlighting the status of the studied diseases in Cambodian goats. 

The objective of the study is clearly articulated. The study design is appropriate to address the stated objectives. The study population is clearly described and appropriate for the hypothesis being tested. The sample size is sufficient to ensure adequate power to address the hypothesis being tested. The statistical analysis used is inadequate to support conclusions. Ethical and regulatory requirements have been met.

However, major corrections are required. The manuscript needs to be strengthened through:

1. Proper use of the English language;

2. Explanation of why the studied diseases can be classified as neglected diseases and also the justification for focusing on them;

3. Use of the most appropriate analysis method;

4. Discussion focused on own findings and implications, including zoonotic ones;

5. Conclusions drawn from the results of the study;

6. Framing of the key recommendations emanating from the findings.

7. Making the figures clear

PLOS authors have the option to publish the peer review history of their article (what does this mean?). If published, this will include your full peer review and any attached files.

Reviewer #1: No

Reviewer #2: No

Reviewer #3: No
---

## [Editor Report · Decision Letter 1]

14 Feb 2023

Dear Dr. Blacksell,

Thank you very much for submitting your manuscript "Risk mapping using serologic surveillance for selected One Health and transboundary diseases in Cambodian goats" for consideration at PLOS Neglected Tropical Diseases. As with all papers reviewed by the journal, your manuscript was reviewed by members of the editorial board and by several independent reviewers. The reviewers appreciated the attention to an important topic. Based on the reviews, we are likely to accept this manuscript for publication, providing that you modify the manuscript according to the review recommendations. 

Sincerely,

Brett M. Forshey

Academic Editor

Dileepa Ediriweera

Section Editor

The authors addressed most of the reviewers' comments. There are only a couple of minor unresolved concerns:

- The Data Availability Statement is still not in line with the PLOS Data Policy. The data need to be made available prior to publication. Simply listing a point of contact is not sufficient, particularly when no contact information is provided. If the data cannot be posted to a public repository, there needs to be a justification, and based on the type of data in this manuscript it's not clear what the restriction would be. Again, the data is supposed to be made publicly available and not just "upon request."

- In the Abstract and elsewhere: The authors did not follow the guidance to focus on seroprevalence to pathogens, not the diseases. In other words, "PPRV" should be used when talking about seroprevalence, not "PPR." Same for FMDV instead of FMD. Please correct throughout, unless when referring to the disease.

Figure Files:

Data Requirements:

Reproducibility:

References

---

## [Editor Report · Decision Letter 2]

14 Mar 2023

Dear Dr. Blacksell,

We are pleased to inform you that your manuscript 'Risk mapping using serologic surveillance for selected One Health and transboundary diseases in Cambodian goats' has been provisionally accepted for publication in PLOS Neglected Tropical Diseases.

Best regards,

Brett M. Forshey

Academic Editor

Dileepa Ediriweera

Section Editor

---

## [Editor Report · Acceptance letter]

29 Mar 2023

Dear Dr. Blacksell,

We are delighted to inform you that your manuscript, "Risk mapping using serologic surveillance for selected One Health and transboundary diseases in Cambodian goats," has been formally accepted for publication in PLOS Neglected Tropical Diseases.

Best regards,

Shaden Kamhawi

co-Editor-in-Chief

Paul Brindley

co-Editor-in-Chief
